# OpenReview forum: "A Differentiable Alignment Framework for Sequence-to-Sequence Modeling via Optimal Transport"
_NeurIPS.cc/2025/Conference — Submitted to NeurIPS 2025_

### Official Review · Reviewer_1PYL · 2025-06-27

**Clarity:** 3
**Significance:** 2
**Originality:** 3
**Rating:** 4
**Confidence:** 4

**Summary:**

This work proposes a differential alignment framework (OTTC) based on the concept of optimal transport. The proposed framework is able to jointly learn the alignment and the classification for ASR in an end-to-end manner. The results indicates that their framework surpass CTC in terms of alignment evaluation, with the peaky behavior being mitigated.

**Questions:**

1. Can the framework perform streaming ASR?
2. What is the result if I just assign uniform distribution on \\(\alpha\\) instead of learning them?

**Ethical Concerns:**

["NO or VERY MINOR ethics concerns only"]

**Final Justification:**

I would like to summarize my final justification below:

* This is an interesting work that has clearly proposed and formulated the OTTC problem and has successfully applied it to a pratical problem (ASR).

* While the idea is intruiguing, the results just adaquetly verify the availability of the method.

Based on the above reasons, I would like to maintain my (favorable) score.

**Limitations:**

yes

**Paper Formatting Concerns:**

not found

**Quality:**

3

**Strengths And Weaknesses:**

**Strength**

* The idea of framing the alignment in ASR towards Optimal Transport (OT) is novel.

* The alignment results are indeed better, and the peaky behavior in CTC-based ASR is mitigated.

**Weakness**

* The ASR results generally lag behind the baseline (CTC).

* There is no explaination about why “the alignments are better, but the classification results are not”.

* The proposed framework is evaluated on only one sequence-to-sequence task (ASR). If the authors could demonstrate its applicability to other tasks, the title would be more appropriate and better aligned with the actual scope of the work, avoiding potential overclaims.

---

> ### Author Response · Authors · 2025-08-01
> **Authors Response to Reviewer 1PYL**
>
> We appreciate the reviewer’s valuable feedback and insightful comments. Below, we address each of the review points in detail.
>
> - **[W1]** *The ASR results generally lag behind the baseline (CTC).*
>
> While the proposed OTTC loss currently trails the CTC in WER, the primary contribution of this work lies in introducing a novel, fully differentiable framework for sequence alignment. This framework directly addresses several longstanding limitations of existing approaches, most notably, the peaky alignment distributions characteristic of CTC-based models.
>
> The presented experiments are intended primarily as a proof of concept to demonstrate the feasibility and potential of the proposed method. We believe that, with further refinement and community engagement, this framework could serve as a foundation for more competitive and computationally efficient ASR systems.
>
> To illustrate its practical relevance, consider the application of real-time subtitle generation, where precise synchronization is paramount. Our results on LibriSpeech 360h show a WER gap that translates to roughly **two additional transcription errors per hour of audio**. For this application, accepting these two minor errors (usually a token deletetion) is a highly acceptable trade-off for ensuring the subtitles are well aligned with the spoken words, rather than having a perfectly transcribed but badly delayed text.
>
> - **[W2]** *There is no explaination about why “the alignments are better, but the classification results are not”.*
>
> This outcome stems from how the models handle timing drift. When the OTTC model's prediction drifts slightly behind the audio, its strict monotonic alignment forces it to re-synchronize, often by deleting a character, which helps maintain accurate alignment. In contrast, CTC's flexibility allows it to use the `blank` token to absorb this drift, preserving the character but sacrificing temporal precision, which results in worse alignment.
>
> - **[W3]** *The proposed framework is evaluated on only one sequence-to-sequence task (ASR). If the authors could demonstrate its applicability to other tasks, the title would be more appropriate and better aligned with the actual scope of the work, avoiding potential overclaims.*
>
> Our primary contribution is the theoretical framework itself, the Sequence Optimal Transport Distance (SOTD), which is defined generally in Section 3 for any sequence type and cost function, without assuming audio.
>
> We chose ASR as a canonical and highly challenging testbed because alignment is a well-known and difficult problem in the field, making it the perfect domain to rigorously validate our framework. Our goal was to provide a deep and thorough validation in one demanding domain to establish a solid foundation for future work, which we hope will explore its application to other seq2seq tasks.
>
> - **[Q1]** *Can the framework perform streaming ASR?*
>
> Yes, the framework is fully compatible with streaming ASR. The Transformer encoder can be made streamable using a standard causal attention mask. During training/inference, the monotonic nature of our alignment allows for a simple, frame-by-frame greedy decoding process suitable for real-time applications. More generally any decoding strategy for CTC works with OTTC.
>
> - **[Q2]** *What is the result if I just assign uniform distribution on $\alpha$ instead of learning them?*
>
> Using fixed and uniform $\alpha$, on the 360-hour LibriSpeech setup with Wav2Vec2-large as the pre-trained model, OTTC yields a WER of 3.51% on test-clean, compared to 2.77% for CTC and 3.00% for OTTC with learnable OT weights. On test-other, the WER was 8.24%, compared to 6.58% for CTC and 7.44% for OTTC with learnable OT weights. This not only degrades ASR performance but, more importantly, it destroys the model's ability to perform meaningful temporal localization, as the mapping from audio to text becomes static and content-agnostic. This result confirms that our learnable alignment mechanism is a critical component of the framework.
>
> We hope we have addressed the reviewer's concerns and would be happy to provide any additional details if needed. We sincerely hope the reviewer will consider revising the scores in light of the clarifications provided.

---

> > ### Comment · Reviewer_1PYL · 2025-08-05
> >
> > Thank you for the detailed response. I have no further questions regarding the work. I would like to summarize my final justification below:
> >
> > * This is an interesting work that has clearly proposed and formulated the OTTC problem and has successfully applied it to a pratical problem (ASR).
> >
> > * While the idea is intruiguing, the results just adaquetly verify the availability of the method.
> >
> > Based on the above reasons, I would like to maintain my (favorable) score.
> >
> > Below are my personal comments and suggestions for the authors; they do not affect the review score:
> >
> > * The observation that “better segmentation does not always lead to better classification results” has been reported elsewhere [1]. I suspect that the *non-causal*, *layered* transformer encoder may introduce ambiguity into the segmental structure of high-dimensional speech features, which could explain why more precise alignment does not necessarily improve classification; and even a uniform distribution on $\alpha$ leads to <1% WER differences.
> >
> > * The claimed advantage of OTTC for real-time subtitle generation appears subjective. Personally, I would not sacrifice classification accuracy for a marginal gain in alignment precision. I recommend not including this claim from the manuscript until a proper human evaluation is conducted. Not to mention that current streaming ASR systems using CTC or RNN-T are not "badly delayed" as the authors imply; they are actually very powerful nowadays.
> >
> > Due to the above reasons, I would like to suggest the authors to not overly focusing on ASR, but on discovering other possible applications. Maybe the significance would be much stronger on a task or domain that does not already have a well-established framework for the alignment-classification problem.
> >
> >
> > ### Reference
> > [1] Tseng, Liang-Hsuan, et al. "REBORN: reinforcement-learned boundary segmentation with iterative training for unsupervised ASR." *NeurIPS*, 2024.

---

> > > ### Author Response · Authors · 2025-08-06
> > > **Authors Response to Reviewer 1PYL**
> > >
> > > Thank you for your thoughtful comments and suggestions. We appreciate your insights on the segmentation/classification trade-off and potential broader applications, and we will examine these points in our future work.

---

### Official Review · Reviewer_Dw35 · 2025-07-02

**Clarity:** 3
**Significance:** 3
**Originality:** 3
**Rating:** 4
**Confidence:** 5

**Summary:**

The paper proposes a differentiable alignment framework for S2S modeling, including a pseudo-metric, called sequence optimal transport Distance (SOTD), a loss for ASR training (Optimal Temporal Transport Classification, OTTC). The experiments on TIMIT, AMI, and Librispeech show the proposed method provides more accurate alignment, although with an increase of the WER.

The paper provides a theoretical framework for the proposed framework, which is interesting.

**Questions:**

Several minor concerns:

1. I thought the authors should have experiments on both TIMIT, AMI, Librispeech for CTC, CR-CTC and the proposed OTTC methods as inferred from Table 1 and Table 2. Why does the alignment results on Librispeech miss in Table 1 and WER results for CR-CTC are missing in Table 2?

2.  The authors mentioned the ablation of training model with the CTC forced alignment. How about a baseline model (maybe it is the top line) trained with the alignment from the hybrid model which more accurate?

3. The paper seems to have no comparison to other alignment model. Accurate alignment is a good to have for the ASR model. If the WER performance is not competitive, it is worth to compare the performance to a pure alignment model.

**Ethical Concerns:**

["NO or VERY MINOR ethics concerns only"]

**Final Justification:**

I will keep my score.

**Limitations:**

The limitation of the paper is the high WER to be used in the practical ASR scenario.

**Quality:**

3

**Strengths And Weaknesses:**

The strengths of the papers are:

1. Novel method and interesting idea.
2. Good performance in terms of the alignment.

The weakness of the paper is:

1. Degraded performance on ASR.

---

> ### Author Response · Authors · 2025-08-01
> **Authors Response to Reviewer Dw35**
>
> We appreciate the reviewer’s valuable feedback and insightful comments. Below, we address each of the review points in detail.
>
>  - **[Q1]** *I thought the authors should have experiments on both TIMIT, AMI, Librispeech for CTC, CR-CTC and the proposed OTTC methods as inferred from Table 1 and Table 2. Why does the alignment results on Librispeech miss in Table 1 and WER results for CR-CTC are missing in Table 2?*
>
> We did not report alignment metrics on LibriSpeech due to the lack of reliable ground-truth alignments. While forced alignments could be generated, their quality is questionable and can be biased by the model used to create them. We chose to report alignment metrics only on datasets where reliable ground-truth (TIMIT) or established protocols for this metric (AMI) exist, to ensure our claims are rigorous.
> CR-CTC was included as a state-of-the-art baseline specifically for **alignment quality**, as it has been shown to mitigate the peaky behavior of CTC. Our goal was to compare OTTC with CR-CTC on this specific front. The standard CTC model served as our main WER baseline.
>
> - **[Q2]** *The authors mentioned the ablation of training model with the CTC forced alignment. How about a baseline model (maybe it is the top line) trained with the alignment from the hybrid model which more accurate? *
>
> A direct comparison using alignments from a hybrid GMM-HMM model is methodologically challenging due to a fundamental mismatch in prediction targets. Hybrid systems are trained on alignments where the neural network predicts acoustic states (pdf-ids). In contrast, our end-to-end OTTC model, like CTC, is trained to directly predict graphemic units such as characters or BPEs. A canonical mapping from the phone-level pdf-id targets of a hybrid system to the BPE-level targets our model requires does not exist, making a direct comparison ill-defined.
>
>
> - **[Q3]** *The paper seems to have no comparison to other alignment model. Accurate alignment is a good to have for the ASR model. If the WER performance is not competitive, it is worth to compare the performance to a pure alignment model.*
>
> We did compare with a “pure” differentiable‑alignment baseline: **soft‑DTW**. The results and discussion are in Appendix B.3 (“Comments on soft‑DTW”). In short, on the 360 h LibriSpeech setup soft‑DTW collapses to trivial paths and yields **39.4 % WER**, far worse than both OTTC (3.00 %) and CTC (2.77 %). This confirms that OTTC achieves much stronger alignment quality and maintains usable recognition accuracy, whereas existing alignment‑only losses do not.
>
> - **Limitations** *The limitation of the paper is the high WER to be used in the practical ASR scenario.*
>
> We argue that the trade-off between alignment quality and transcription accuracy should be viewed in the context of the intended application. For a significant class of problems, superior alignment is the most critical requirement, and a modest increase in WER is an acceptable compromise for the substantial gains in temporal precision that our method provides.
> For instance, in domains like **medical speech analysis**, the main goal is often to pinpoint the exact temporal location of pathological cues, such as stuttering or voice disorders. In this context, accurate alignment is more critical than a marginal difference in the transcribed text. Likewise, for **automatic subtitle generation**, especially for live content, ensuring that subtitles are perfectly synchronized with the spoken words is essential for user experience and accessibility. Misaligned subtitles can be more disruptive than a minor transcription error.
> Therefore, while the standard CTC model may be preferable for tasks where only the final transcription is valued, our OTTC framework offers a compelling and robust solution for these alignment-critical applications. Our work provides a new, effective tool for these important scenarios, and we believe future research can build upon this foundation to reduce the WER gap, potentially offering the best of both worlds.
>
> We hope we have addressed the reviewer's concerns and would be happy to provide any additional details if needed. We sincerely hope the reviewer will consider revising the scores in light of the clarifications provided.

---

### Official Review · Reviewer_gpaD · 2025-07-03

**Clarity:** 3
**Significance:** 3
**Originality:** 3
**Rating:** 5
**Confidence:** 3

**Summary:**

This paper presents the SOTD training objective, which can be applied to the seq2seq tasks like ASR. Compared with previous training objectives like CTC and CR-CTC, the proposed method shows less peaky behavior and smoother alignment prediction.

The SOTD is rooted in the optimal transport theory. It assumes the input and output as two series of one-hot distributions, based on which the author defines the Sequences Optimal Transport Distance (SOTD). The authors also compared the proposed method and previous CTC and DTW methods.

**Questions:**

About the time and space complexity: The authors claim that the proposed method has linear time and space complexity over the seq2seq task, which should be a very nice property.

I would appreciate it if the authors could add a comparison of the complexity between the proposed method and CTC. If you can show that the proposed method has better time and space complexity, I guess it would be widely applied to some extreme cases. E.g., for the super-long seq2seq tasks, like long-form ASR, the CTC would have complexity more than linear, which makes it infeasible to fit the GPU memory and becomes slow. If the proposed method is theoretically more suitable for this situation, I would encourage the authors to address this slightly.

**Ethical Concerns:**

["NO or VERY MINOR ethics concerns only"]

**Final Justification:**

The paper provides a new training objective for sequence-to-sequence tasks. With a solid theoretical foundation and primary but comprehensive experiments, the merits of the proposed approach are well-defended. The proposed method even has a linear perplexity, which is beneficial for super-long alignment training, a good nature that is unseen in previous CTC.

**Limitations:**

No noticiable limitation

**Paper Formatting Concerns:**

No formatting concern

**Quality:**

4

**Strengths And Weaknesses:**

Strengths:
(1) The paper has a good theoretical foundation. The reviewer hasn't found a noticeable flaw so far, although the reviewer may not have enough knowledge to fully check the correctness of all theoretical part.
(2) The paper conducted enough experiments on TIMIT, AMI, and Librispeech, showing its ASR and alignment performance. For a theoretical paper, this experimental scale is enough (although further scaling could be more convincing, it's not necessary)

Weakness:
(1) The proposed method shows a little bit worse performance, compared with CTC, as shown in Table 2. As an early exploration, this is acceptable.

---

> ### Author Response · Authors · 2025-08-01
> **Authors Response to Reviewer gpaD**
>
> We appreciate the reviewer’s positive response and valuable feedback.
>
> This is a key advantage of our method: our alignment cost is linear in sequence length. As stated in Section 4.2, for input length $n$ and label length $m$, OTTC computes the 1D OT coupling in $O(max(n,m))$ time and memory because the bins are already sorted and the 1D transport is solved with a single forward pass without constructing a full cost matrix.
> By contrast, the standard CTC forward–backward recursion is $O(nm)$ time and typically $O(nm)$ memory. The gap between the two methods can be witnessed experimentally. We benchmarked forward+backward runtimes (median over 10 runs, batch size 64, $m = 0.3n$) on a single GPU :
> | $n$ (frames) | OTTC (s) | CTC-native (s) |
> |------------------|----------|----------------|
> | 1060 (~21s) | 0.0007   | 0.0028     |
> | 2060             | 0.0010   | 0.0093     |
> | 3060             | 0.0013   | 0.0160     |
> | 4060             | 0.0015   | 0.0290     |
> | 5060             | 0.0018   | 0.0398     |
> | 6060             | 0.0021   | 0.0601     |
> | 7060             | 0.0024   | 0.0869     |
> | 8060             | 0.0027   | 0.1037     |
> | 9060             | 0.0030   | 0.1348     |
> | 10060(~210s)  | 0.0033   | 0.1581     |
>
> A logarithmic least-squares fit gives an exponent of $≈0.7$ for OTTC and $≈1.8$ for CTC, consistent with near-linear vs. near-quadratic growth. Even with our OTTC prototype PyTorch implementation, OTTC’s computation time is already low; a low level implementation should further reduce it.
>
> **Implications for long-form ASR.** When $n$ reaches tens of thousands (e.g., meetings, audiobooks), CTC’s $O(nm)$ DP becomes slow and memory-heavy, whereas OTTC still scales linearly. This makes OTTC particularly attractive for extreme-length seq2seq settings.

---

> > ### Comment · Reviewer_gpaD · 2025-08-08
> >
> > Thanks for the reply. I really appreciate the insight the reviewer provided during the rebuttal. The paper is of good quality and merits acceptance.
> >
> > (Maybe it's a bit late, and whether the authors answer this or not would not negatively affect my judgement) Have you tried to compare the convergence speed of the proposed method and CTC? Say, given the same amount of computing or steps, would the proposed method achieve similar performance to CTC, or which one would need more steps/compute?
> >
> > best,

---

> > > ### Author Response · Authors · 2025-08-08
> > > **Authors Response to Reviewer gpaD**
> > >
> > > Thanks for your reply and positive feedback.
> > >
> > > We did not explicitly compare the convergence speed of the proposed OTTC and CTC models. However, in our experiments, we trained both models for 40 epochs. The CTC model generally achieved its best validation score around epochs 28–29, whereas OTTC usually reached its best validation score around epochs 37–38. At the same time, we observed that the alignment in OTTC converged much faster, which motivated our decision to fix the alignment sub-modules after 30 epochs.

---

### Official Review · Reviewer_xwcA · 2025-07-04

**Clarity:** 4
**Significance:** 3
**Originality:** 4
**Rating:** 5
**Confidence:** 4

**Summary:**

This paper introduces a method based on optimal transport to jointly optimize alignment and transcription for the ASR task, which does not involve marginalizing across alignments. The goal of the proposed method is to obtain more accurate alignments and avoid the peaky behavior exhibited by CTC.
Experiments on TIMIT, AMI and Librispeech show that the proposed method obtains more accurate alignments and avoid the peaky behavior. On the ASR task, the proposed method still lags behind CTC but not substantially.

**Questions:**

Problem formulation seems to assume the same dimension d for input vectors and output vectors but x_i has dimension d while y_i has dimension L (1-hot vector).

WER is higher for the proposed method: 1) a qualitative analysis could inform better whether the difference is meaningful 2) it would be interesting to confirm the difference on a large-scale and noisier dataset like VoxPopuli

Have you considered force aligning Librispeech in order to also report alignment metrics on Librispeech as well (even though the paper claims this would not be reliable)?

**Ethical Concerns:**

["NO or VERY MINOR ethics concerns only"]

**Final Justification:**

This is in my opinion a solid paper and the authors provided a detailed rebuttal answer so I will maintain my score of 5.

**Limitations:**

Yes

**Quality:**

4

**Strengths And Weaknesses:**

Strengths:
* extremely well written paper, with clear mathematical presentation
* good motivation for the work
* experimental results are promising

Weaknesses:
* the proposed method lags on the ASR task

---

> ### Author Response · Authors · 2025-08-01
> **Authors Response to Reviewer xwcA**
>
> We appreciate the reviewer’s valuable feedback and insightful comments. Below, we address each of the review points in detail.
>
> - **[Q1]** *Problem formulation seems to assume the same dimension d for input vectors and output vectors but x_i has dimension d while y_i has dimension L (1-hot vector).*
>
> The function $F$ maps each acoustic frames $x_i \in R^d$ to a logit vector in $R^{|L|+1}$. The cross-entropy cost is then computed between the resulting probability distribution and the one-hot target, ensuring dimensional consistency.
>
> - **[Q2]** *WER is higher for the proposed method: 1) a qualitative analysis could inform better whether the difference is meaningful*
>
> OTTC shares many errors with CTC. Both models typically make the **same** acoustic mistakes, for example, confusing “chock” with “shock.” Beyond these overlaps, most of OTTC’s additional errors are deletions. Visual inspection of the alignments shows that when the model’s predictions drift slightly behind the audio, OTTC often skips a character entirely to re‑synchronize and avoid misalignment for the rest of the sentence.
>
> This behavior arises from fundamental differences in how the two models handle timing drift. CTC’s architecture is inherently flexible: by marginalizing over all paths, it can use the blank token to absorb small timing delays without penalty, making it robust to local misalignments. In contrast, OTTC enforces a single, strict monotonic alignment and cannot use blanks as a buffer. When it falls behind, the alignment cost rapidly accumulates, prompting the model to delete a character to minimize this penalty and catch up. This higher deletion rate is the chief source of the WER gap and the price paid for the superior, non‑peaky alignment metrics our method achieves.
>
> - **[Q3]** *2) it would be interesting to confirm the difference on a large-scale and noisier dataset like VoxPopuli*
>
> We thank the reviewer and agree that an evaluation on VoxPopuli would further strengthen our work. Our current results already demonstrate strong performance in terms of both scalability and noise robustness.
>
> **Scalability**: Our LibriSpeech experiments (from 100h to 960h) show that OTTC scales consistently with data size, achieving similar relative improvements over CTC at each scale.
>
> **Noise robustness**: On the challenging AMI corpus, OTTC still produces significantly better alignments than CTC (IDR 42.84 vs. 16.75), although the improvement is smaller than in the clean TIMIT setting (IDR 76.72 vs. 26.98).
>
> We can’t run VoxPopuli within the rebuttal time frame, but we will run it and commit to including the full results in the appendix in the final version of the paper.
>
> - **[Q4]** *Have you considered force aligning Librispeech in order to also report alignment metrics on Librispeech as well (even though the paper claims this would not be reliable)?*
>
> We did not report alignment metrics on LibriSpeech due to the lack of reliable ground-truth alignments. While forced alignments could be generated, their quality is questionable and can be biased by the model used to create them. We chose to report alignment metrics only on datasets where reliable ground-truth (TIMIT) or established protocols for this metric (AMI) exist, to ensure our claims are rigorous.

---

> > ### Comment · Reviewer_xwcA · 2025-08-04
> >
> > Thanks for the detailed response. I don't have further questions at this point.
> >
> > Best,
> >
> > -Reviewer

---

### Decision · Program_Chairs · 2025-09-17

**Decision:**

Reject

**Comment:**

The paper proposed Optimal Temporal Transport Classification (OTTC) that jointly learn alignment and performance classification in an end-to-end manner, specifically addressing the "peaky behavior" and alignment inaccuracies of established methods like CTC.

The reviewers agree with the novel application of optimal transport for the alignment problem and the paper is well presented. The experimental results successfully demonstrated the effectiveness of the proposed method in terms of better alignments.

The primary and consistent concern raised by all reviewers is the trade-off between alignment and transcription accuracy. The proposed OTTC method consistently lags behind baseline CTC in terms of WER.
The authors argued that "For a significant class of problems, superior alignment is the most critical requirement, and a modest increase in WER is an acceptable compromise for the substantial gains in temporal precision that our method provides." and took "medical speech analysis" and "automatic subtitle generation" as examples. However, from my speech industry experience, transcription accuracy always goes before alignment accuracy.

The paper only compared against CTC, while in the ASR field, RNN-T is another popular and wide adopted streaming speech recognition model (also mentioned by 1PYL). It's important to include comparisons with it.

Another important data point is to compare against the forced-alignment from hybrid ASR models. The authors argued that "forced alignments could be generated, their quality is questionable" and "comparison using alignments from a hybrid GMM-HMM model is methodologically challenging". In practice, forced alignment from hybrid ASR models have been used as ground truth to build almost all phoneme-based ASR systems, the quality of those alignment would not be a problem. Taking a step back, if the quality is indeed bad as the authors said, wouldn't it be an extra point to show case the benefit of the proposed method.
Secondly to the point of comparison between phoneme-based alignment model and grapheme based ones, a simple approach is to convert both alignment to word-level and compare differences at word-level, which is the actual metrics matters in real world speech applications.

Based on the above concerns, I hence recommend rejection despite of the reviewers' scores (two accept and two borderline accept).